# Unbiased and Signal-Weakening Photoelectrochemical Hexavalent Chromium Sensing via a CuO Film Photocathode

**DOI:** 10.3390/nano13091479

**Published:** 2023-04-26

**Authors:** Wenxiang Lu, Lu Ma, Shengchen Ke, Rouxi Zhang, Weijian Zhu, Linling Qin, Shaolong Wu

**Affiliations:** School of Optoelectronic Science and Engineering, Collaborative Innovation Center of Suzhou Nano Science and Technology, Key Lab of Advanced Optical Manufacturing Technologies of Jiangsu Province, Key Lab of Modern Optical Technologies of Education Ministry of China, Soochow University, Suzhou 215006, China

**Keywords:** self-powered sensing, signal weakening, photoelectrochemical sensor, hexavalent chromium, CuO film

## Abstract

Photoelectrochemical (PEC) sensors show great potential for the detection of heavy metal ions because of their low background noise, high sensitivity, and ease of integration. However, the detection limit is relatively high for hexavalent chromium (Cr(VI)) monitoring in addition to the requirement of an external bias. Herein, a CuO film is readily synthesized as the photoactive material via reactive sputtering and thermal annealing in the construction of a PEC sensing photocathode for Cr(VI) monitoring. A different mechanism (i.e., Signal-Weakening PEC sensing) is confirmed by examining the electrochemical impedance and photocurrent response of different CuO film photoelectrodes prepared with the same conditions in contact with various solutions containing concentration-varying Cr(VI) for different durations. The detection of Cr(VI) is successfully achieved with the Signal-Weakening PEC response; a drop of photocathode signal with an increasing Cr(VI) concentration from the steric hindrance effect of the in situ formed Cr(OH)_3_ precipitates. The photocurrent of the optimized CuO film photocathode linearly declines as the concentration of Cr(VI) increases from 0.08 to 20 µM, with a detection limit down to 2.8 nM (Signal/Noise = 3) and a fitted sensitivity of 4.22 µA·μM^−1^. Moreover, this proposed sensing route shows operation simplicity, satisfactory selectivity, and reproducibility.

## 1. Introduction

With the advancement of industry, global environmental pollution has become more severe. Heavy metal ions are one of the most important pollutants in water pollution, affecting all parts of the environment, such as terrestrial and aquatic communities. Heavy metal ions have high toxicity and enrichment and are difficult to degrade. Hexavalent chromium ion (Cr(VI)) is one of the most harmful pollutants to human health and the environment [1]. General speaking, the toxicity of chromium elements is largely related to its valence state. Cr(VI) is well known as a carcinogen and an extremely toxic inhaled substance. Excessive exposure to Cr(VI) can lead to excessive accumulation in human tissues, which has been associated with diseases such as bronchogenic carcinoma, asthmas, pneumonitis, and dermatitis [1,2]. The environmental problems caused by Cr(VI) cannot be ignored. However, trivalent chromium ion Cr(III) is an important component of glucose tolerance factor in the human body. An appropriate amount of Cr(III) can help insulin promote the efficiency of glucose entering cells, and is an important blood glucose regulator [3]. The guideline for Cr(VI) content prescribed by the World Health Organization (WHO) in drinking water is 0.96 µM [4]. However, Cr(VI) is used extensively in industrial processes such as electroplating, tanning, textile dyeing, corrosion inhibition, and wood treatment. Industrial processes and products have led to widespread chromium contamination in the environment. Today, increasing attention is being paid to solving the problem of chromium wastewater pollution [5,6,7]. Therefore, it is essential to develop a novel sensor that can effectively detect low-concentration Cr(VI) in industrial wastewater and ideally reduce Cr(VI) to Cr(III).

Currently, a variety of analytical techniques are available for Cr(VI) detection, including atomic absorption spectrometry [8], fluorescence [9], UV-vis spectrophotometry [10], inductively coupled plasma mass spectrometry [11], surface-enhanced Raman spectroscopy [12], and ion chromatography [13]. Although these methods have some advantages, the required equipment is expensive and bulky, which is inconvenient for popularization and promotion, and is not suitable for rapid detection of large-scale on-site samples [14]. Recently, electrochemical (EC) and photoelectrochemical (PEC) methods, because of their simplicity, fast detection, low cost, ease of miniaturization, and on-site analysis, have been successfully used in the detection of various analytes [15,16]. In particularly, PEC analysis is a novel and unique sensing method with relatively high sensitivity and a relatively low detection limit because of the separation of excitation and detection signals. PEC sensors can be irradiated with ubiquitous solar light as an excitation signal, and convert the optical signal into an electrical signal via the photoelectronic effect of the photoactive material [17]. The PEC and spectroscopic sensing methods have different application scenarios. Spectroscopic sensing relies on large instruments with very high sensitivity, but they are only applicable to the professional laboratory and testing organization. The portable PEC sensors show great potential for the detection of heavy metal ions because of their low background noise, ease of integration, and suitability for rapid detection of large-scale on-site samples. It should be noted that the performance of PEC sensors depends largely on the properties of photoactive materials.

In recent years, most of the PEC sensors for sensing Cr(VI) have employed a metal oxide with excellent photoelectrochemical properties as the photoactive material. For instance, Siavash Moakhar et al. proposed a screen-printed TiO_2_ nanoparticle photoanode modified with gold nanoparticles and achieved a high sensitivity (up to 11.88 µA·μM^−1^) of Cr(VI) PEC detection, with a wide detectable range (0.01 μM–100 µM) and a detection limit of 0.004 µM at a reverse bias of 0.2 V [18]. It should be pointed out that the reported PEC sensor based on a screen-printed TiO_2_ nanoparticle photoanode shows an anodic photocurrent response and a signal enhancement with the increasing Cr(VI) concentration. Li et al. constructed the Bi surface plasmon resonance (SPR)-promoted BiPO_4_/BiOI heterostructures for a cathodic-signal-enhanced PEC sensor and realized the Cr(VI) detection with a linear range of 0.5–180 µM and a limit of detection (LOD) of 0.15 µM at a bias of −0.2 V [19]. However, most of the current PEC sensors for Cr(VI) determination require a certain working voltage, and the LOD is relatively high. P-type CuO material, with high catalytic activity and great sunlight absorption capacity, is cheap, environmentally friendly, and rich in natural resources. Due to the above excellent chemical and physical properties, increasing numbers of developments and applications for the CuO material, such as CO_2_ photocatalytic reduction [20], photoelectrochemical water splitting [21], photocatalytic reduction of Cr(VI) [22], fast adsorption of noxious Cr(VI) ions from the solvent phase [23], and glucose monitoring [24], have been developed. As a material with good light absorption and excellent photoelectric activity, CuO has rarely been used for PEC sensing of Cr(VI).

In this work, we achieve an unbiased PEC Cr(VI) sensor simply by coating a CuO film on an FTO conductive glass with reactive sputtering and post-thermal annealing. Through the electrochemical impedance and photocurrent responses of the different CuO film photocathodes in contact with various solutions containing concentration-varying Cr(VI) for different durations, the Signal-Weakening sensing mechanism for Cr(VI) detection is demonstrated. Additionally, the optimized PEC sensor can enable a wide range of Cr(VI) detection with high sensitivity, ultra-low detection limit, and good selectivity at relatively zero bias.

## 2. Materials and Methods

### 2.1. Materials and Reagents

Cu sputtering target was purchased from Zhongnuo Advanced Material (Beijing, China) Technology Co., Ltd. FTO conductive glass was purchased from commercial sources in China. Analytical-grade potassium dichromate (K_2_Cr_2_O_7_), a source of Cr(VI), was purchased from Yonghua Chemical Co., Ltd. (Suzhou, China). Tris phosphine hydrochloride (tris HCl) was purchased from Phygene Co., Ltd. (Fuzhou, China). Zinc chloride (ZnCl_2_), potassium chloride (KCl), ferric chloride (FeCl_3_), and chromic nitrate (Cr(NO_3_)_3_) were purchased from Macklin Co., Ltd. (Shanghai, China). Sodium chloride (NaCl) and Copper sulfate (CuSO_4_) were purchased from Aladdin Co., Ltd. (Shanghai, China). Potassium hexacyanoferrate (K_3_[Fe(CN)_6_]) and potassium ferrocyanide (K_4_[Fe(CN)_6_]) were purchased from Shanghai Guoyao Co., Ltd. (Shanghai, China). Deionized water with an electrical resistivity of >18.2 MΩ·cm was used. These chemicals were used as obtained without further purification.

### 2.2. Preparation of the CuO Film Photoelectrode

CuO coating was performed by reactive sputtering (LN-1082FS, Kejing, Shenyang, China) of metallic Cu target followed by thermal annealing. The substrate for the reactive sputtering process was cleaned FTO glass. The deposition power was 80 W, the deposition pressure was 0.8 Pa, the gas flow ratio of Ar to O_2_ (Ar/O_2_) was 40/10 sccm, and the deposition time was optimized to 30 min. Thermal treatment was performed on the as-obtained CuO films at 500 °C for 1 h in air according to optimization results.

The CuO films were prepared to be photoelectrodes with the following steps. First, a layer of indium was soldered on the front side of the FTO where there was no film coverage, then the Cu conductive adhesive was used to fix the Cu wires to the substrate containing indium, and finally the back, side walls, and part of the front side of the photocathode were wrapped with silicone rubber and insulating waterproof adhesive, leaving an active area with a diameter of 4 mm. Note that the CuO film photoelectrodes used in this work were all disposable.

### 2.3. Material Characterizations

Microstructure and morphology were characterized by field emission scanning electron microscopy (SEM, Zeiss, Sigma 300, Jena, Germany). Optical absorbance spectra were obtained from a UV-Vis-NIR spectrophotometer with a 150 mm InGaAs integrating sphere (PerkinElmer, Lambda 1050S+, Shanghai, China). X-ray powder diffraction (XRD, Bruker D8 Advance, Karlsruher, Germany) was used to analyze the crystalline phase with the phase identification determined by using Cu Kα radiation. X-ray photoelectron spectroscopy (XPS, Thermo Fisher, ESCALab 250 Xi, MA, USA) with a monochromatic Al Kα source (1486.7 eV) was used to analyze the valence states of material elements at an operational pressure of 10^−8^ mBar, and the XPS spectrum was calibrated with the peak of carbon contamination C1 (284.8 eV). Inductively coupled plasma mass spectrometry (ICP-MS, ICAP RQ, MA, USA) was used to analyze the concentration of chromium element in the solution used for PEC measurements.

### 2.4. Photoelectrochemical Measurements

PEC sensing performance and analysis were carried out using a three-electrode electrochemistry workstation (Zennium, Zahner, Kronach, Germany). The three-electrode configuration was equipped with the as-prepared CuO film photoelectrode as the working electrode (WE), the Pt mesh as the counter electrode (CE), and the saturated calomel electrode (SCE) as the reference electrode (RE). The employed light source was simulated AM 1.5G illumination (Enlitech, S-F7-3A, Taiwan, China) with a calibrated power density of 100 mW·cm^−2^, and the 10 mM tris HCl (pH = 7) containing different and concentration-varying analytes was placed in a home-made quartz photolysis cell as the analytical fluid. The photocurrent response was measured under 60-sec-ON/60-sec-OFF circular illumination. Chronoamperometry at zero bias relative to the RE under the ON-OFF circular AM 1.5G illumination was used to examine the photocurrent responses, sensitivity, selectivity, and repeatability. Under AM 1.5G illumination, K_2_Cr_2_O_7_, KCl, CuSO_4_, NaCl, Cr(NO_3_)_3_, FeCl_3_, ZnCl_2_, and K_2_Cr_2_O_7_ were dripped successively on the background solution (i.e., 10 mM tris HCl solution) to assess sensing selectivity. The volumes of interfering ion solutions added dropwise were fixed to be 1 mL, which was successively mixed with the background solution. The final concentrations of K_2_Cr_2_O_7_, KCl, CuSO_4_, NaCl, Cr(NO_3_)_3_, FeCl_3_, ZnCl_2_, and K_2_Cr_2_O_7_ in the selectivity testing were 1.64 μM, 15.87 μM, 15.63 μM, 15.38 μM, 14.71 μM, 14.49 μM, 14.28 μM, and 1.41 μM after mixing with the background solution. The magnetic stirrer is in constant rotation throughout the selectivity test. Electrochemical impedance spectroscopy (EIS) was carried out in the same three-electrode configuration and 0.1 M KCl solution mixing with 1 mM K_4_[Fe(CN)_6_] and 1 mM K_3_[Fe(CN)_6_]. EIS was recorded at open-circuit potential from 100 kHz to 0.01 Hz with an amplitude of 5 mV under AM 1.5G illumination.

## 3. Results and Discussion

The morphology of the CuO films deposited on FTO glass substrates is shown in Figure 1a. One can see that a rough film is successively deposited. The rougher surface provides more active sites for PEC sensing. In order to determine the thickness of the CuO film more accurately, we used the same conditions to deposit CuO film on the planar Si substrate as a contrast. Figure 1b shows that the CuO thickness is ~160 nm when the sputtering time is 30 min. XPS analysis was used to confirm the surface elements and their chemical states. Figure 1c shows two main Cu 2p XPS peaks with the binding energy of 933.4 eV and 953.6 eV, respectively, attributed to Cu 2p_3/2_ and Cu 2p_1/2_; meanwhile, the satellite peaks (around 941.09 eV, 943.8 eV, and 961.9 eV) are all consistent with the Cu^2+^ in CuO [25,26]. The binding energy associated with the O 1s XPS spectrum in Figure 1d shows a strong peak at 529.28 eV, which is associated with metal-oxygen bonding in the CuO lattice. Meanwhile, the peak around 530.84 eV is an impurity peak which exhibits hydroxyl oxygen on the surface [27]. The XPS spectra were also examined on the different samples with different heat treatment temperatures, but the peak differences were trivial. This proves that the coated film with a uniform thickness and rough surface is copper oxide.

As shown in the XRD spectra of the bare FTO substrate and the CuO films deposited on the FTO substrates with different annealing temperatures (Figure 1e), the five peaks are marked by the star belonging to the FTO substrate. The XRD patterns display that some diffractions of the CuO films annealed at different temperatures in the 2θ range of 20–80° are identified with pure CuO polycrystalline phase (PDF#45-0937). With an increase in the annealing temperature, the intensities of the peaks corresponding to CuO(110), (111), (200), and (202) increase, indicating that the crystallinity of the CuO film is improving [28]. Specifically, the CuO(200) peak was not found for the unannealed sample. This indicates that the annealing temperature has a significant effect on the crystallinity. For the main XRD peak of CuO(111), the crystal size can be calculated from the XRD peak intensity. We calculated the crystal size and found the higher the annealing temperature, the larger the crystal size (which is in the range of 15–45 nm). The absorption spectra (Figure 1f) show that CuO film photoelectrodes have a broad absorption in visible and ultraviolet bands, and the annealing treatment leads to a slight increase of optical absorption in the whole wavelength range of interest, which can be explained by the band gap of CuO being narrower than that of Cu_2_O [28]. This facilitates the use of broadband-spectrum light (e.g., sunlight) and implies the anticipated PEC performance of the CuO film photoelectrode.

According to the above experimental phenomenon, the temperature is demonstrated to have non-negligible effects on the crystallinity and optical properties of copper oxide film [29]. We further optimized the annealing temperature for the PEC sensing photoelectrode. Photocurrent response curves and EIS were used to investigate the PEC activity of copper oxide film photoelectrode. Figure 2 shows that the cathodic photocurrent of the sample annealed at 500 °C is the largest, and the sample without annealing treatment has the smallest photocurrent, indicating that the photoactive material is a p-type semiconductor, and all the samples are photocathodes. After the photocurrent response measurement, these photocathodes were immediately transferred to 0.1 M KCl solution mixed with 1 mM K_4_[Fe(CN)_6_] and 1 mM K_3_[Fe(CN)_6_] to analyze the carrier transport dynamic. Figure 2b shows that the charge transfer resistance for the sample annealed at 500 °C is the smallest. The relationship of the transfer resistance agrees well with that of the photocurrent. The improvement in crystal quality can result in the reduction of defect density and enhancement of carrier concentration for the CuO film annealed at higher temperatures. Here, the heat treatment at 600 °C leading to a photocurrent smaller than that with heat treatment at 500 °C is related to the fact that the FTO substrate cannot withstand very high temperatures (e.g., 600 °C) [30]. Therefore, the annealing treatment temperature was optimized to be 500 °C for the following sensing measurements.

Considering the substantial PEC activity of the CuO film photocathodes, we further validate the feasibility of PEC Cr(VI) sensing. Photocurrent response (i.e., current versus time) curves of different CuO film photocathodes prepared with the same conditions were recorded. The first group of three CuO film photocathodes was dividually and continuously tested in 10 mM tris HCl solution containing 1 µM K_2_Cr_2_O_7_ for 40 min, 60 min, and 80 min. As shown in Figure 3a, the photocurrent response slowly and steadily decreased when increasing the testing time. After the photocurrent response measurement, the photoelectrodes were immediately placed in 0.1 M KCl solution mixed with 1 mM K_4_[Fe(CN)_6_] and 1 mM K_3_[Fe(CN)_6_] to obtain the EIS. The employed equivalent circuit model is shown in the inset of Figure 3d. The Nyquist impedance plots contain a semicircle at high frequencies, corresponding to the chemical capacitance of CuO and charge transfer resistance (*R*_ct_), and the start of the semicircle indicates the series resistance (*R*_s_). Figure 3d shows that the longer testing time, the greater charge transfer resistance is present. We preliminarily suspect that this is closely related to the surface adsorption of Cr(VI) [23].

Influences of immersion time in solution on the observed photocurrent are investigated to reveal the potential adsorption reaction. The second group of three CuO film photocathodes is immersed in 10 mM tris HCl solution containing 1 µM K_2_Cr_2_O_7_ for various periods in the dark before conducting the photocurrent response measurements. As shown in Figure 3b, pre-immersion has a significant impact on the photocurrent response, that is the photocurrent becomes smaller and smaller as the pre-immersion time increases. In subsequent EIS analysis (Figure 3e), the case with the longer pre-immersion time has the larger charge transfer resistance. These results prove that the surfaces of the CuO film photocathodes do adsorb some ions or molecules in contact with the solution no matter whether the PEC measurement is carried out or not, resulting in increased *R*_ct_ and then decreased photocurrent. Since our aim was to sense Cr(VI), we further prepared the third group of different CuO film photocathodes, whose PEC responses were examined in a 10 mM tris HCl solution containing concentration-varying Cr(VI) without pre-immersion treatment. As shown in Figure 3c, the photocurrent decreased regularly with an increase in the Cr(VI) concentration, indicating that the CuO film photocathode can be used to determine Cr(VI) concentration. After measuring the photocurrent responses, the EIS results in Figure 3f show that the semicircle in the EIS Nyquist plot (i.e., transfer resistance) is smaller for the sample after being measured in the lower concentration of Cr(VI).

The fit results with the equivalent circuit model for the EIS in Figure 3d–f are summarized in Table 1. The longer time taken for the CuO film photocathode to contact with the K_2_Cr_2_O_7_ solution leads to the larger *R*_ct_, and the greater *R*_ct_ is also obtained when the CuO film photocathode is measured in the higher concentration of K_2_Cr_2_O_7_. However, the series resistance shows a trivial and ruleless change. Therefore, it can be concluded that the CuO film photoelectrode can achieve Cr(VI) concentration monitoring with the cathodic Signal-Weakening mode, which is different from many other sensing strategies for PEC Cr(VI) sensing [31,32,33].

In order to confirm the sensing feasibility and reveal the sensing mechanism for PEC Cr(VI) sensing with the CuO film photocathode, we analyzed the surface composition of the different samples after being tested in different solutions. As illustrated in Figure 4a, when the Cr(VI) concentration gradually increases from 0.08 µM to 80 µM, the overall photocurrent at the same measuring time steadily decreases. The corresponding Cr 2p XPS peak in Figure 4b could be fitted into two main peaks: the one located at lower binding energy (573.1 eV) corresponds to Cr(III), and the other peak centered at higher binding energy (575.4 eV) originates from Cr(VI) [34,35]. It is obvious that the area ratio of the Cr(III) sub-peak to the total peak for the sample being tested in the higher-concentration Cr(VI) is larger than that in the lower-concentration Cr(VI), confirming the presence of more Cr(III) in the photocathode surface for the Cr(VI) solution with a higher concentration. Prior to PEC analysis, the nominal concentrations of the solutions were 2.0 µM and 0.2 µM, and the measured solution concentrations from inductively coupled plasma mass spectrometry (ICP–MS) were 1.916 µM and 0.208 µM, respectively. Following photocurrent response testing for the two photocathodes in the 10 mM tris HCl containing 2.0 µM and 0.2 µM Cr(VI), after the 10 min test, the chromium concentrations in the used solution decreased to 1.592 µM and 0.126 µM, respectively. We believe that there are two chemical reactions on the surface of the CuO film photocathode. The first one is the reduction of Cr(VI) to Cr(III), and the next reaction occurs when Cr(III) combines with the hydroxide in the solution to form Cr(OH)_3_. Upon illumination, photogenerated electrons are excited to the conduction band, producing positively-charged holes in the valence band of CuO, and the photogenerated electrons assist in reducing Cr(VI) to Cr(III) if they do not recombine. Related processes can be expressed by Equations (1–3):(1)$$\mathrm{CuO}+h\nu \to \mathrm{CuO}+h^{+}+e^{-}$$
(2)$${\mathrm{Cr}}_{2}{O_{7}}^{2-}+14H^{+}+6e^{-}\to 2{\mathrm{Cr}}^{3+}+7H_{2}O$$
(3)$${\mathrm{Cr}}^{3+}+3{\mathrm{OH}}^{-}\to \mathrm{Cr}{\left(\mathrm{OH}\right)}_{3}$$

As we all know, the reaction in Equation (2) implies an increase in the cathodic photocurrent as the Cr_2_O_7_^2−^ concentration increases. However, the steric hindrance effect is introduced from the formation of Cr(OH)_3_ adsorbed on the CuO film surfaces [36]. Therefore, the cathodic photocurrent decreases in Figure 4a can be explained by the steric hindrance effect of the Cr(OH)_3_ precipitate, which prevents the photogenerated electrons to reduce Cr(VI).

The thorough sensing performance of the CuO film photocathode is finally characterized. Figure 5a presents the cathodic photocurrent responses of the different CuO film photocathodes prepared with the same conditions, and the photocurrent obviously decreases with an increase in the Cr(VI) concentration. In contrast, all the observed dark currents are smaller than 20 nA. The response and recovery time for our prepared sensor is less than 2 sec, and the response time is limited by the switching time of the employed light source. We used the average value of the relatively stable photocurrents in the last 30 s to fit the sensitivity. Figure 5b shows the corresponding derived calibration curves with a fitted sensitivity of 4.22 µA·μM^−1^ in the range of 0.08–2 µM and another fitted sensitivity of 0.18 µA·μM^−1^ in the range of 2–20 µM, and the LOD was calculated to be 2.8 nM (Signal/Noise = 3). Thus, the suggested protocol can be successfully employed for the monitoring of Cr(VI) in drinking water, as the detection requirement of 0.96 µM set by the WHO is included in the linear detection range and two orders of magnitude higher than the detection limit. As a control, the similar PEC and EC detections of Cr(VI) in the previous literature are summarized in Table 2. The comparison shows that our prepared PEC sensor based on the CuO film photocathode has the lowest LOD and upper-level sensitivity. It is worth mentioning that the Cr(VI) sensing can be operated without an external bias, which is also the root cause for the small noise current and LOD.

Selectivity and reproducibility are also key elements for designing a new sensor. The reproducibility is examined by fabricating tens of CuO film photoelectrodes with the same conditions and performing the amperometric experiments in background solutions with and without 1 µM K_2_Cr_2_O_7_. The averaged photocurrents for each sample are shown in Figure 5c, indicating that the photocurrent deviation for the two solutions is within ±15%. Anti-interference measures were performed by successively adding foreign ions into the background solution. As is shown in Figure 5d, a significant drop of the photocurrent is present when K_2_Cr_2_O_7_ is injected into the background solution, but no obvious changes are observed from injecting a wide range of foreign ions, including KCl, CuSO_4_, NaCl, Cr(NO_3_)_3_, FeCl_3_, and ZnCl_2_. This phenomenon is mainly due to the fact that the position of the conduction band of CuO is more negative than the reduction potential of Cr(VI), and more positive than the reduction potential of Cr(III), so the accumulated electrons in CuO can reduce Cr(VI) to Cr(III). Additionally, Cr(VI) has a greater redox potential than other substances. As a result, the accumulated electrons in CuO can selectively reduce Cr(VI) to Cr(III) [37]. The results indicate that our prepared sensor possesses satisfactory repeatability and selectivity.

**Table 2 nanomaterials-13-01479-t002:** A comparison of Cr(VI) detection performance with different working electrodes.

Sensing Component	Test Conditions	Sensitivity (µA μM^−1^)	Linear Range (µM)	LOD (µM)	Method	Ref.
FTO/TiO_2_/Au NPs photoanode	−0.2 V ^b^ Simulated sunlight	11.88	0.01–100	0.004	PEC Signal enhancing	[18]
Bi/BiPO_4_/BiOI photocathode	−0.2 V ^b^ Visible light	0.315 0.111	0.5–10 10–180	0.3	PEC Signal enhancing	[19]
FTO/TiO_2_ NRs/Au NPs photoanode	−0.2 V ^b^ Simulated sunlight	13.94	0.01–50	0.006	PEC Signal enhancing	[31]
BiOI/CN photoanode	0.3 V ^b^ Visible light	0.0027 0.0003	0.5–15 15–190	0.1	PEC Signal enhancing	[32]
ITO/MoS_2_/BiOI photocathode	0 V ^b^ Visible light	0.3174 0.0388	0.05–10 10–160	0.01	PEC Signal enhancing	[33]
Ti/TiO_2_ NTA/NiCoLDHs photoanode	0 V ^a^ UV light	2.15 lg*C* 6.50 lg*C* 1.10 lg*C*	0.5–20 20–40 400–1800	0.12	PEC Signal weakening	[38]
BiVO_4_ photoanode	0.4 V Visible light	0.123 log*C*	2–10	0.01	PEC Signal enhancing	[39]
Ti/TiO_2_ NTs/Au NPs electrode	0.28 V ^a^	6.91	0.1–105	0.03	EC	[40]
GCE/Ag film electrode	0.59 V ^a^	1.590	0.35–40	0.1	EC	[41]
MWCNTs/Au NPs electrode	0.52 V ^b^	0.28	0.8–230	0.72	EC	[42]
FTO/CuO film photocathode	Rel 0 V ^b^ Simulated sunlight	4.22	0.08–2	0.0028	PEC Signal weakening	This work

^a^: vs. Ag/AgCl. ^b^: vs. SCE. NTAs: Nanotube arrays. NiCo-LDHs: Nickel-cobalt layered double hydroxides. NRs: Nanorods. NPs: Nanoparticles. NTs: Nanotubes. MWCNTs: Multi-walled carbon nanotubes.

## 4. Conclusions

In summary, a PEC Cr(VI) sensor based on a CuO film photocathode was conveniently prepared using reactive sputtering and post-annealing treatment. The signal-weaking mechanism for the Cr(VI) detection was demonstrated; that is, the steric hindrance effect of the Cr(OH)_3_ precipitate in situ formed during the PEC Cr(VI) sensing process. The as-prepared PEC Cr(VI) sensor can work simply under solar light illumination without an external bias, and possesses a detection limit down to 2.8 nM and a sensitivity of 4.22 µA·μM^−1^ in the concentration range of 0.08–2 µM, as well as satisfactory reproducibility and anti-interference. This work provides an alternate strategy for the self-powered and trace detection of heavy metal ions, and sheds light on the new applications for CuO-based photoactive materials.

## Figures and Tables

**Figure 1 nanomaterials-13-01479-f001:**
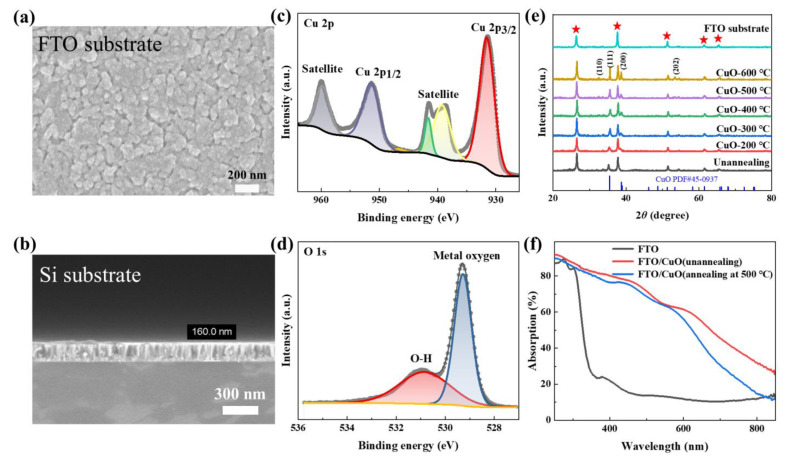
(**a**) Top-view SEM image of the CuO film coated on the FTO substrate. (**b**) Sectional-view SEM image of the CuO film coated on the planar Si substrate. (**c**,**d**) Cu 2p and O 1s XPS peaks were recorded from the surface of the as-annealed CuO film photocathode. (**e**) XRD spectra of the FTO substrate and CuO film photocathodes annealed at different temperatures. (**f**) Absorption spectra of the bare FTO glass substrate, and annealed and unannealed CuO films coated on the FTO substrate.

**Figure 2 nanomaterials-13-01479-f002:**
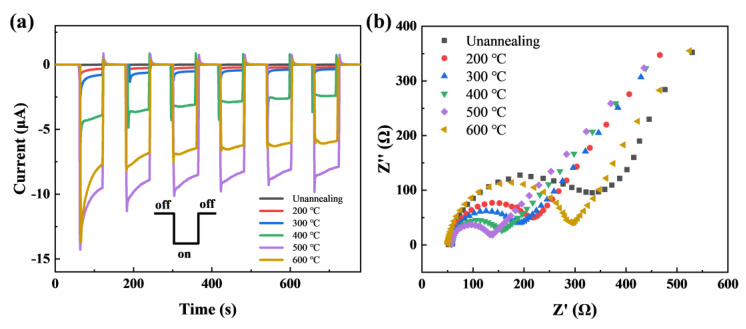
(**a**) Photocurrent response curves for the CuO film photocathodes annealed at different temperatures under ON-OFF circular AM 1.5G illumination in the 10 mM tris HCl solution. (**b**) EIS curves of these photoelectrodes in (**a**) obtained in 0.1 M KCl solution mixing with 1 mM K_4_[Fe(CN)_6_] and 1 mM K_3_[Fe(CN)_6_].

**Figure 3 nanomaterials-13-01479-f003:**
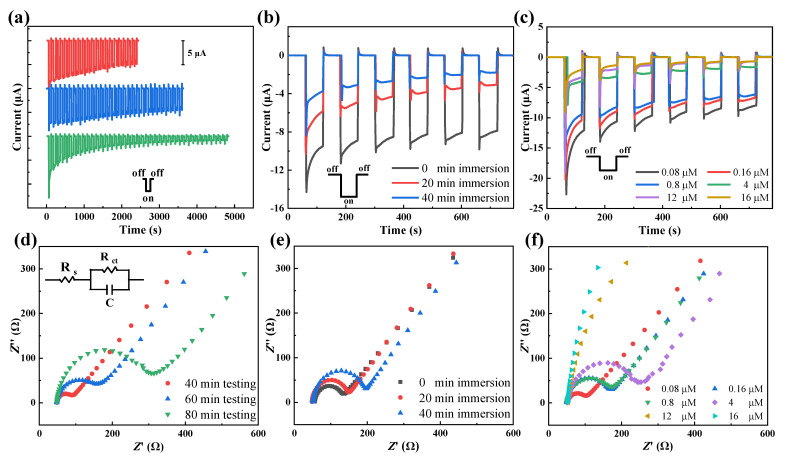
(**a**) Photocurrent response curves for the different CuO film photocathodes continuously tested for different duration times. (**b**) Photocurrent response curves for the different photocathodes after being immersed in 1 µM K_2_Cr_2_O_7_ solution for different times. The used electrolyte in (**a**,**b**) is the 10 mM tris HCl containing 1 µM K_2_Cr_2_O_7_. (**c**) Photocurrent response curves for the different photocathodes in the 10 mM tris HCl containing concentration-varying Cr(VI). (**d**) EIS for the samples after measurement in (**a**). (**e**) EIS for the samples after measurement in (**b**). (**f**) EIS of these samples after measurement in (**c**). The used solution for EIS analysis is 0.1 M KCl solution mixed with 1 mM K_4_[Fe(CN)_6_] and 1 mM K_3_[Fe(CN)_6_].

**Figure 4 nanomaterials-13-01479-f004:**
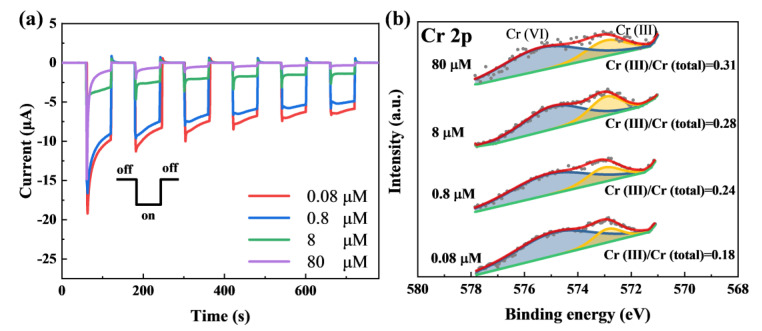
(**a**) Photocurrent response of the different CuO film photocathodes measured in the background solution containing concentration-varying Cr(VI). (**b**) Cr 2p XPS peaks of the photocathodes in (**a**) after PEC measurements.

**Figure 5 nanomaterials-13-01479-f005:**
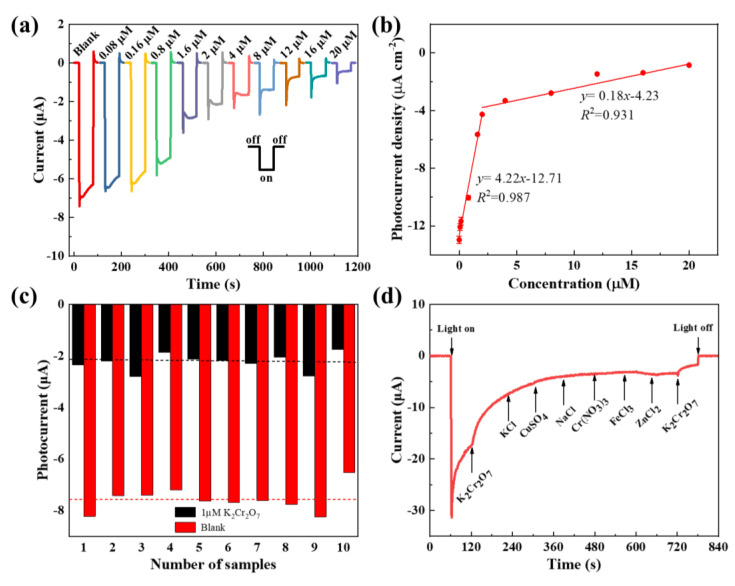
(**a**) Photocurrent responses of the different CuO film photocathodes in the background solution containing concentration-varying Cr(VI). (**b**) Calibration curves versus Cr(VI) concentration. (**c**) Repeatability evaluation of the photocurrent of the CuO film photocathodes in the background solutions with or without 1 µM K_2_Cr_2_O_7_. The horizontal dash corresponds to the averaged photocurrent. (**d**) Selectivity evaluation of the photocurrent of the CuO film photocathode.

**Table 1 nanomaterials-13-01479-t001:** The fitting charge transfer resistance obtained from the Nyquist impedance plots.

		Testing Time (min)			Immersion Time (min)			Cr(VI) Concentration (µM)					
		40	60	80	0	20	40	0.08	0.16	0.8	4	12	16
*R*_s_ (Ω)		47.58	46.95	47.55	55.16	47.6	50.25	53.04	47.89	49.7	51.9	52.86	50.51
*R*_ct_ (Ω)		55.46	134.6	277.4	85.81	109.5	156.1	92.74	131.8	136.2	206.4	968.5	1917

## Data Availability

The data presented in this study are available on request from the corresponding author.

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
