# Peer review of "Unbiased and Signal-Weakening Photoelectrochemical Hexavalent Chromium Sensing via a CuO Film Photocathode"

_nanomaterials, 2023, doi:10.3390/nano13091479_

Round 1

Reviewer 1 Report

The authors have demonstrated the development of an unbiased photoelectrochemical (PEC) sensor for detecting Cr(VI) using a CuO film photocathode. This sensor was conveniently prepared through reactive sputtering and post-annealing treatment. The signal weakening PEC mechanism was confirmed through various experiments, including electrochemical impedance and photocurrent response from different CuO film photoelectrodes. The authors were able to successfully designed the Cr(VI) sensor with a detection limit of 2.8 nM. The manuscript is well-written. Please find my comments,

1.      Regarding selectivity, in line 305, the authors did not mention how the sensor is selective for Cr(VI). It is unclear if they tested the CuO film for sensing other substances and concluded that it is selective only to Cr(VI).

2.      What will be the lifetime of the CuO film and is it reusable.

3.      Is it possible to compare the PEC sensor created by the authors and detection methods such as Raman or surface-enhanced Raman spectroscopy?.

Reviewer 2 Report

In the manuscript authors reported the synthesis of CuO film by reactive sputtering and thermal annealing for the photoelectrochemical sensing of hexavalent chromium (Cr(VI)). The concept is scientifically sound and quite interesting. However, there are some defects which need to be corrected. Thereby, I would recommend the manuscript for publication in Nanomaterials after major revision. Some comments are provided as follows:

  1. It is recommended to estimate the optical band gap of the materials using Tauc Plot based on UV-Vis DRS results. 

  2. Maybe the authors should carry out XPS analysis not only of the as-annealed film, but also of the samples annealed at various temperature.

  3. Authors should specify what procedure for evaluating the crystallinity from XRD patterns is used and calculated the average particle size.

  4. Generally characterization section needs more improvement by analysis of the basic materials.

  5. Authors claim that a 500 °C is the most suitable temperature for the film annealing and obtaining maximum photocurrents. However, there is no explanation why a sample annealed at a given temperature performs best. This fact should be explained.

  6. Figure 4a should also provide the photocurrent response of the CuO film photocathode measured in the background solution without Cr(VI) containing .

  7. Response and recovery time are found to be the important factors in the sensing properties. Thereby, it is recommended to evaluate these parameters during the performance of the prepared CuO film.

  8. The long-term stability of catalysts is found to be the important factor in the PEC and EC performance. Thereby, it is recommended to study and evaluate this parameter during the experiments using standard protocols.

  9. For a better understanding of differences in photoelectrochemical performance of samples, the electrochemical active surface area (ECSA) should be estimated and presented.

  10. The text is not free from grammatical and punctuation errors. Please ask a native English speaker to revise and proofread their revised manuscript before re-submission.

  11. The authors may note several papers related to this work were published in the Nanomaterials. Hence, they are strongly advised to cite some of the recent references from this journal in the revised version of the manuscript.

Reviewer 3 Report

The authors present a paper about the photoelectrochemical sensing of Cr(VI) by a CuO photocathode. This studi is in principle interesting, but I have serious doubts that the reported results are meaningful for the following reasons:

1 - The authors are stating that interference by other analites like Cu2+, Hg2+, Ag+, etc was tested. They also say that the have used HgSO4, AgCl and PbSO4 as source for Pb2+, Ag+ and Hg2+. All these compounds are insoluble salts, or are not stable in water like HgSO4 (insoluble Hg3O2SO4 is formed)! How can they use 1000microM solutions? (page 3, lines 142-147)

2 - The also write that they test each group of CuO photocathode in 0.1 mM (tris?) HCl solution. Cuo is an alkaline oxide and as such is quickly dissolved in HCl according to the following reaction: CuO + 2HCl(aq) --> CuCl2(aq) + H2O. So, how can this be? CuO is soluble in an acidic solution! Maybe the decrease in photocurrent they detect is simply due to a progressive dissolution of the oxide.

I think that the whole experimental design and analysis needs to be reconsidered. 

Round 2

Reviewer 3 Report

The authors have partly corrected the issues raised in my previous version. The tris HCl buffer solution should be called with the proper name: Tris(2-carboxyethyl)phosphine hydrochloride. With regard to the insoluble salts used to see the interference (AgCl, etc.) they should write that the concentration is the one calculated from the solubility product
